# Noise Reduction Method of Underwater Acoustic Signals Based on CEEMDAN, Effort-To-Compress Complexity, Refined Composite Multiscale Dispersion Entropy and Wavelet Threshold Denoising

**DOI:** 10.3390/e21010011

**Published:** 2018-12-24

**Authors:** Guohui Li, Qianru Guan, Hong Yang

**Affiliations:** School of Electronic Engineering, Xi’an University of Posts and Telecommunications, Xi’an 710121, China

**Keywords:** underwater acoustic signals, CEEMDAN, effort-to-compress complexity, refined composite multiscale dispersion entropy, wavelet threshold denoising

## Abstract

Owing to the problems that imperfect decomposition process of empirical mode decomposition (EMD) denoising algorithm and poor self-adaptability, it will be extremely difficult to reduce the noise of signal. In this paper, a noise reduction method of underwater acoustic signal denoising based on complete ensemble empirical mode decomposition with adaptive noise (CEEMDAN), effort-to-compress complexity (ETC), refined composite multiscale dispersion entropy (RCMDE) and wavelet threshold denoising is proposed. Firstly, the original signal is decomposed into several IMFs by CEEMDAN and noise IMFs can be identified according to the ETC of IMFs. Then, calculating the RCMDE of remaining IMFs, these IMFs are divided into three kinds of IMFs by RCMDE, namely noise-dominant IMFs, real signal-dominant IMFs, real IMFs. Finally, noise IMFs are removed, wavelet soft threshold denoising is applied to noise-dominant IMFs and real signal-dominant IMFs. The denoised signal can be obtained by combining the real IMFs with the denoised IMFs after wavelet soft threshold denoising. Chaotic signals with different signal-to-noise ratio (SNR) are used for denoising experiments by comparing with EMD_MSE_WSTD and EEMD_DE_WSTD, it shows that the proposed algorithm has higher SNR and smaller root mean square error (RMSE). In order to further verify the effectiveness of the proposed method, which is applied to noise reduction of real underwater acoustic signals. The results show that the denoised underwater acoustic signals not only eliminate noise interference also restore the topological structure of the chaotic attractors more clearly, which lays a foundation for the further processing of underwater acoustic signals.

## 1. Introduction

The underwater acoustic signals processing is one of the most active subjects in modern information fields [1,2]. Underwater acoustic signal is a non-linear, non-Gaussian, non-stationary chaotic signal, it is easily effected by the other targets, marine environment and various equipment in the process of acquisition and transmission. These problems will inevitably cause some noise for received signal, which is extremely harmful to target detection, location, classification and recognition [3,4]. And self-characteristic of signal is easily ignored in the traditional underwater acoustic signals denoising methods. Therefore, the more effective technology has a wide application prospect in underwater acoustic signal processing [5,6,7]. 

Empirical Mode Decomposition (EMD) is proposed by Huang [8], it is an adaptive decomposition method for processing some non-linear and non-stationary signals. EMD not only does not need to set a basis function but also can overcome the shortcomings of subjective experience. However, mode mixing and boundary effects can easily occur in the decomposition process, so that the different intrinsic mode functions will contain similar components which affect the decomposition effect [9]. In order to overcome these difficulties, Wu and Huang [10] proposed a noise-assisted analysis method ensemble empirical mode decomposition (EEMD). EEMD is the improvement of EMD, the mode mixing can be basically eliminated. However, some residual noise exist in the reconstructed components, the calculation amount of EEMD is larger than EMD. Besides, empirical mode decomposition energy entropy has received largely extensive attention, such as the field of roller bearing fault diagnosis [11].

A complete ensemble empirical mode decomposition with adaptive white noise (CEEMDAN) [12] was proposed. This method can obtain the IMFs by adding adaptive white noise and calculating specific allowance based on EEMD. CEEMDAN can solve these problems that imperfect decomposition process and larger reconstruction error of EMD. Moreover, it can require a few iterations, which can save a lot of computational costs. At present, the CEEMDAN has been widely applied to non-linear signal processing, such as power load prediction [13], gear fault diagnosis [14], medical signal processing [15], wind speed prediction [16] and so forth. 

In order to further research the characteristic information of non-linear and non-stationary signals, many methods for measuring complexity have been proposed. such as sample entropy (SampEn) [17], permutation entropy (PE) [18,19], approximation entropy [20], multi-scale permutation entropy (MPE) [21,22] and multi-scale sample entropy (MSE) [23] and so forth. However, for chaotic time series, the calculating speed of SampEn is slower and the mutated signal is more changeable. Although the calculating speed of PE is faster than SampEn, the relationship between signal amplitudes is easily neglected. The application of FE in big data processing is quite difficult because the fuzzy measure of signal cannot be determined. In response to the inherent flaws of the above methods, a new complexity measurement method, dispersion entropy (DE) [24], which was proposed by Rostaghi and Azami in 2016. This method has the advantages of the faster calculating speed and better stability, moreover, it is never affected by mutation signals. Inspired by multi-scale entropy and composite multi-scale, Azami made an improvement of DE and multiscale dispersion entropy (MDE), proposed a refined composite multiscale dispersion entropy (RCMDE) [25]. Compared with other multiscale methods, RCMDE has the advantages of better stability, faster calculation speed, stronger stability and higher recognition rate and so forth. And it is more suitable for researching and processing the non-linear and non-stationary signal. The RCMDE has been applied to check diagnosis of rolling bearings [26]. 

In view of the advantages of CEEMDAN and RCMDE in non-linear dynamics, they are applied to non-linear chaotic signals and actual underwater acoustic signals and proposed a noise reduction algorithm combined with ETC and wavelet threshold denoising. The key points of the proposed algorithm are as follows: (1) the IMFs by CEEMDAN are reorganized into four reconstructed IMFs, which is beneficial to signal denoising. (2) Wavelet soft threshold denoising is applied to recombination series, the optimal decomposition level is determined by the noise reduction effect of reconstructed IMFs. (3) The proposed algorithm is respectively applied to Chens model and actual underwater acoustic signals. (4) Through qualitative and quantitative analysis to denoised signal, the noise reduction effect of the proposed algorithm is verified by comparing with EMD_MSE_WSTD and EEMD_DE_WSTD.

## 2. Basic Theory

### 2.1. Complete Ensemble Empirical Mode Decomposition with Adaptive Noise (CEEMDAN) 

#### 2.1.1. EEMD

EEMD is an algorithm based on EMD, the specific steps of EEMD are summarized as follows [27]:

Step 1: Define original signal is x(t), $\varepsilon^{i}\left(t\right)$ represents white noise series with standard normal distribution in the i-th experiment. $x^{i}\left(t\right)=x\left(t\right)+\varepsilon^{i}\left(t\right)$ is defined as the i-th signal, where i=1,…,N represents the number of experiments. 

Step 2: We can obtain $IMF_{n}^{i}$ by decomposing $\varepsilon^{i}\left(t\right)$ from EMD, where n=1,…,K is the number of IMFs. 

Step 3: The n-th modal component is defined as $IMF_{n}$, the average value of $IMF_{n}^{i}$ is expressed as:(1)$$IMF_{n}=\frac{1}{N}\sum_{i}^{N}IMF_{n}^{i}$$

Because of white noise $\varepsilon^{i}\left(t\right)$ involved in the decomposition in each experiment is different, so the residual signals are different. The residual signals are defined as:(2)$$r_{n}^{i}\left(t\right)=r_{n-1}^{i}-IMF_{n}^{i}\left(t\right)$$

#### 2.1.2. CEEMDAN

CEEMDAN is an improved algorithm of EEMD. The key technique of this algorithm is that adds adaptive white noise and calculates the specific residuals to obtain corresponding IMFs. $E_{n}\left(*\right)$ is defined as the n-th modal component by EMD. In this paper, $\bar{\bar{IMF_{n}}}\left(t\right)$ represents the n-th modal component obtained by CEEMDAN. the specific steps of CEEMDAN are summarized as follows [28]:

Step 1: Perform N experiments on the signal $x^{i}\left(t\right)$, the first modal $\bar{\bar{IMF_{1}}}\left(t\right)$ is defined as:(3)$$\bar{\bar{IMF_{1}}}\left(t\right)=\frac{1}{N}\sum_{i=1}^{N}IMF_{1}^{i}\left(t\right)$$

Step 2: Calculate the first residual, $r_{1}\left(t\right)=x\left(t\right)-\bar{\bar{IMF_{1}}}\left(t\right)$. 

Step 3: The i(i=1,…,N) experiments are conducted continuously $r_{1}\left(t\right)+E_{1}\left(\varepsilon^{i}\left(t\right)\right)$ is decomposed until the first modal component of EMD is obtained. Calculate the second modal component $\bar{\bar{IMF_{2}}}\left(t\right)$:(4)$$\bar{\bar{IMF_{2}}}\left(t\right)=\frac{1}{N}\sum_{i=1}^{N}E_{1}\left(r_{1}\left(t\right)+E_{1}\left(\varepsilon^{i}\left(t\right)\right)\right)$$

Step 4: Calculate the n-th residual signal and the (n+1)-th modal component for the remaining phases according to the calculation process of step 3.
(5)$$r_{n}\left(t\right)=r_{n-1}\left(t\right)-\bar{\bar{IMF_{n}}}\left(t\right)$$
(6)$$\bar{\bar{IMF_{\left(n+1\right)}}}\left(t\right)=\frac{1}{N}\sum_{i=1}^{N}E_{1}\left(r_{n}\left(t\right)+E_{n}\left(\varepsilon^{i}\left(t\right)\right)\right)$$

Step 5: Continue to execute step 4 until the residual signal is no longer decomposed this criterion is that the number of extreme points of residual signal is less than or equal to 2. the final residual signal is as follows:(7)$$R\left(t\right)=x\left(t\right)-\sum_{n=1}^{K}\bar{\bar{IMF_{n}}}\left(t\right)$$

Therefore, original signal will be eventually decomposed as:(8)$$x\left(t\right)=\sum_{n=1}^{K}\bar{\bar{IMF_{n}}}\left(t\right)+R\left(t\right)$$

This decomposition process is complete and adaptive from the algorithm implementation steps of CEEMDAN. This method not only accurately reorganize original signal by adding different white noise also restore some characteristics of EMD while solving the mode mixing problem. The flow chart of CEEMDAN [29] is designed in Figure 1.

### 2.2. Effort-To-Compress Complexity (ETC)

Shannon entropy has been widely used to characterize the complexity of time series caused by stochastic processes such as the chaotic dynamic systems. However, it does not perform well on noise-intensive non-stationary time series. By the time, the measure of compression complexity becomes a great alternative solution [30]. At present, the two methods have been used to measure the complexity of signals, namely Lempel-Ziv test (LZ) and effort-to-compress test (ETC). Many researches have confirmed that LZ and ETC are superior to Shannon entropy in accurately characterizing the dynamic complexity of non-linear dynamic systems. However, the ETC has more different complexity value than LZ and Shannon entropy, it can achieve more sophisticated resolution [31]. Therefore, the application of ETC in non-linear, non-stationary underwater acoustic signals should be promising. 

ETC is used to compress a known series by an algorithm, which named non-sequential recursive pair substitution (NSRPS) [32], the specific steps of ETC are summarized as follows:

Step 1: Input time series is converted to symbol series so as to compress them by compression calculation, where $s_{1},\ldots ,s_{n}$ are defined as input series with the number of n. 

Step 2: The first iteration is performed when the number of a symbol occurrences is greatest, the series will be replaced by a new symbol. For example, the symbol of “11010010” can be transformed into “12202” caused by “10” has the largest number of occurrences compared with “00”, “01” and “11” in the iteration. 

Step 3: Complete the second iteration continuously, in which “12202” is converted to “3202”. In fact, the frequency of occurrence of all symbols is no difference, we choose to replace “12”.

Step 4: The remaining series $s_{2},\ldots ,s_{n}$ are respectively iterated according to above algorithm until the length of string is reduced to 1 or the series becomes a constant series. Thus, the change of series “11010010” is expressed as: 11010010→12202→3202→402→52→6. 

Step 5: The value of complexity by ETC is obtained according to execution times of this algorithm. where p is defined as the value of ETC, N represents the number of algorithm that required to convert an input series into a constant series by NSRPS.
(9)$$p=\frac{N}{L-1}$$
where *L* represents the length of symbol series, N is a non-negative integer from 0 to L−1, attention should also be paid to:(10)$$0\leq \frac{N}{L-1}\leq 1$$

### 2.3. Refined Composite Multiscale Dispersion Entropy (RCMDE)

#### 2.3.1. Dispersion Entropy (DE)

DE is a non-linear dynamic analysis method that characterizes the complexity and irregularity of time series, the algorithm is based on the mapping of normal distribution function. Thus, the expectation and standard deviation of data should be considered, the calculation steps of DE are summarized as follows [33]:

Step 1: Define time series is $x=\left\{x_{j},j=1,2,\ldots ,N\right\}$, x is mapped to $y=\left\{y_{j},j=1,2,\ldots ,N\right\}$ according to normal distribution function, where $y_{j}\in \left(0,1\right)$. and the normal distribution function $y_{j}$ is defined as:(11)$$y_{j}=\frac{1}{\sigma \sqrt{2\pi }}\int_{-\infty }^{x_{j}}e^{\frac{-{\left(t-\mu \right)}^{2}}{2\sigma^{2}}}dt$$
where μ and σ respectively represent expectation and standard deviation of time series. 

Step 2: The y is mapped to the range of [1,2,…,c] by linear transformation.
(12)zjc=R(c·yj+12)
where R and c respectively represent integer function and the number of categories. 

Step 3: Calculate the embedded vector $z_{i}^{m,c}$:(13)$$z_{i}^{m,c}=\left\{z_{i}^{c},z_{i+d}^{c},\ldots ,z_{i+\left(m-1\right)d}^{c}\right\}$$
where i=1,2,…,N−(m−1)d, m and d respectively represent embedding dimensions and time delays. 

Step 4: The dispersion pattern is defined as: $\pi_{v_{0},v_{1},\ldots ,v_{m-1}}\left(v=1,2,\ldots ,c\right)$,if $z_{i}^{c}=v_{0},z_{i+d}^{c}=v_{1},\ldots ,z_{i+\left(m-1\right)d}^{c}=v_{m-1}$, the dispersion patterns of $z_{i}^{m,c}$ is $\pi_{v_{0},v_{1},\ldots ,v_{m-1}}$. 

Step 5: For each dispersion patterns, relative frequency is defined as follows:(14)$$p\left(\pi_{v_{0},v_{1},\ldots ,v_{m-1}}\right)=\frac{N_{b}\left(\pi_{v_{0},v_{1},\ldots ,v_{m-1}}\right)}{N-(m-1)d}$$
where $N_{b}\left(\pi_{v_{0},v_{1},\ldots ,v_{m-1}}\right)$ represents the number of $z_{i}^{m,c}$ mapped to $\pi_{v_{0},v_{1},\ldots ,v_{m-1}}$. Actually, $p\left(\pi_{v_{0},v_{1},\ldots ,v_{m-1}}\right)$ shows the ratio of $N_{b}\left(\pi_{v_{0},v_{1},\ldots ,v_{m-1}}\right)$ to $z_{i}^{m,c}$. 

Step 6: According to the definition of Shannon, the DE of original time series is defined as:(15)$$DE\left(x,m,c,d\right)=-\sum_{\pi =1}^{c^{m}}p\left(\pi_{v_{0},v_{1},\ldots ,v_{m-1}}\right)\ln\left(p\left(\pi_{v_{0},v_{1},\ldots ,v_{m-1}}\right)\right)$$

The parameters selection has been described in Reference [34]. the value of embedding dimensions and categories should be appropriate, m is usually taken as 2 or 3, c is taken as an integer from 3 to 8, d is 1, the length of input time series should be greater than 2000. 

From the results of DE, the larger the value of DE, the higher the irregularity of time series, conversely, the irregularity is lower. From the process of algorithm, when all possible dispersion patterns obtain equal probability value, the complexity of signal is the highest, DE obtains the maximum value $\ln\left(c^{m}\right)$. If there is a value of $p\left(\pi_{v_{0},v_{1},\ldots ,v_{m-1}}\right)$ and it is not zero, it shows that the lower the complexity of time series, the smaller the value of DE [35].

#### 2.3.2. Multiscale Dispersion Entropy (MDE)

MDE is an interval method based on DE for measuring the complexity and regularity of time series. and the specific steps of MDE are summarized as follows [36,37]:

Step 1: For initial time series x, when embedding dimension m and similar tolerance r are respectively determined, the k-th coarse-grained time series with the scale factor τ can be constructed. $w_{k}^{\tau }$ is defined as:(16)$$w_{k}^{\left(\tau \right)}=\sum_{a=\left(j-1\right)\tau +1}^{j\tau }x_{a},\;k=1,2,\ldots ,N/\tau$$
where τ is a positive integer. For each scale factor, sample data is divided into several sequences with length N/τ. 

Step 2: The MDE with the change of scale factor can be obtained according to the coarse-grained time series, the DE of sample data for each scale factor is defined as: (17)$$MDE\left(x,m,c,d,\tau \right)=\frac{1}{\tau }\sum_{k=1}^{\tau }DE\left(w_{k}^{\tau },m,c,d\right)$$

The MDE algorithm fulfills multiscale transformation by equidistant segmentation and averaging of original data. Although its calculation process is simple and fast, there are some relationships between the data after segmentation, which can easily lead to the loss of information. 

#### 2.3.3. Refined Composite Multiscale Dispersion Entropy (RCMDE)

In order to solve the above problems, original data is pre-processed on the basis of MDE and then improve the algorithmic process. the specific steps of RCMDE are summarized as follows:

Step 1: For original time series x, length(∗) represents the length of signal. the k-th of coarse-grained series is $w_{k}^{\tau }=\left\{w_{k,1}^{\tau },w_{k,2}^{\tau },\ldots ,w_{k,j}^{\tau },\ldots ,w_{k,N}^{\tau }\right\}$, where k=1,2,…,τ, $w_{k,j}^{\tau }$ is defined as:(18)$$w_{k,j}^{\tau }=\frac{1}{\tau }\sum_{a=k+\tau \left(j-1\right)}^{k+j\tau -1}x_{a},\;1\leq j\leq \frac{length\left(x\right)}{\tau }$$

Step 2: For each scale factor, RCMDE is defined as follows:(19)$$RCMDE\left(x,m,c,d,\tau \right)=-\sum_{\pi =1}^{c^{m}}\bar{p}\left(\pi_{v_{0},v_{1},\ldots ,v_{m-1}}\right)\ln\left(\bar{p}\left(\pi_{v_{0},v_{1},\ldots ,v_{m-1}}\right)\right)$$
where $\bar{p}\left(\pi_{v_{0},v_{1},\ldots ,v_{m-1}}\right)$ represents the average probability of dispersion pattern $w_{k}^{\tau }$, $\bar{p}\left(\pi_{v_{0},v_{1},\ldots ,v_{m-1}}\right)$ is defined as follows:(20)$$\bar{p}\left(\pi_{v_{0},v_{1},\ldots ,v_{m-1}}\right)=\frac{1}{\tau }\sum_{1}^{\tau }p_{k}^{\tau }$$

In this paper, the relevant characteristics of RCMDE is verified by analyzing the simulated signals. And the mean value and standard deviation figures of Gaussian white noise (Noise 1) and 1/f noise (Noise 2) with the length of 3000 are plotted. These figures respectively reflect the mean value and standard deviation of MSE, DE and RCMDE in 10 scale factors. The parameters of MSE are set to: embedding dimension m=2, similar tolerance r=0.15,and the parameters of MSE are defined as follows: m=2, c=3, d=1. The simulation results of Noise 1 and Noise 2 are shown in Figure 2.

As can be seen from Figure 2, the overall trend of this three methods is basically the same. It is found that the entropy of Noise 1 is larger than that of Noise 2 on the low scale, the entropy of Noise 1 monotonously decreases with the increase of scale. Which indicates that the degree of irregularity of Gaussian noise is higher, the main information is on the low scale. However, the curve change of Noise 2 is not obvious, which indicates that internal structure of Noise 2 is more complex, its main information is not on the low scale. As shown in Figure 2c, the curve of RCMDE is smoother and more stable. Therefore, it has been verified that the RCMDE is more suitable for analyzing these two signals, the stability and accuracy are higher.

### 2.4. Wavelet Threshold Denoising

An effective signal noise reduction method can play a vital role in the field of signal processing. Wavelet analysis developed from Fourier analysis is a new time-frequency analysis tool, which has favorable time-frequency localized and multi-resolution properties. Wavelet analysis has been widely applied in signal processing field [38,39,40]. The specific steps of wavelet transform are as follows:

We suppose that the mathematical expression of one dimensional signal with noise is f(t)=s(t)+δe(t), where t=0,1,…,(n−1). s(t), e(t), f(t) and δ are respectively defined as real signal, Gaussian noise, noise signal and the correlation coefficient of noise. 

Step 1: A proper wavelet basis function and decomposition level are selected to perform wavelet decomposition on the noisy signal f(t). 

Step 2: For the high frequency coefficients obtained by wavelet decomposition, the thresholds are estimated according to the appropriate threshold selection criteria. 

Step 3: The high frequency coefficients in different decomposition scales are quantified by thresholds. 

Step 4: The low frequency coefficients of wavelet decomposition and high frequency coefficients after processing are reconstructed to obtain denoised signals. 

There are many wavelet basis functions and threshold selection criteria in the wavelet analysis, the db4 wavelet basis function is used in this paper. Owing to soft threshold denoising can flexibly overcome the discontinuity of hard threshold among many threshold estimation methods, which has been widely applied to signal processing fields. Therefore, the wavelet soft threshold denoising (WSTD) is applied to this paper.

## 3. The Proposed Noise Reduction Algorithm

### 3.1. The Proposed Noise Reduction Algorithm

In this paper, a noise reduction method based on CEEMDAN, effort-to-compress complexity, refined composite multiscale dispersion entropy and wavelet soft threshold denoising is proposed. The flow chart of the proposed algorithm is designed in Figure 3.

The specific steps of the proposed algorithm are as follows:

Step 1: An input signal is decomposed into several IMFs by CEEMDAN and arranged from high frequency to low frequency in turn. 

Step 2: Calculating the ETC of all IMFs. If the ETC is greater than or equal to threshold p, this IMF will be determined as noise IMF. And set p as 0. 85 after multiple experiments.

Step 3: Calculating the RCMDE of remaining IMFs. If the RCMDE is greater than or equal to threshold q, this IMF is defined as noise-dominant IMFs. If the RCMDE is greater than or equal to threshold r, this IMF is the real signal-dominant IMFs. and the remaining IMFs are judged as the real IMFs, where q is 1.85, r is 1.10.

Step 4: Wavelet soft threshold denoising is applied to noise-dominant IMFs and real signal-dominant IMFs. Owing to the noise signals contain different degrees of noise, the optimal denoising effect is distributed in different decomposition levels. In this paper, wavelet basis function is db4, decomposition level is from one to six, the optimal decomposition level is determined according to signal noise ratio (SNR) and root mean square error(RMSE) of denoised IMFs. 

Step 5: In order to ensure more effective denoising effect, the noise IMFs are abandoned. We can obtain the final denoised signal by combining real IMFs and denoised IMFs obtained by wavelet soft threshold.

In steps 2 and 3, the algorithm of determining *p*, *q*, *r* is as follows:

Step 1: Many researches have confirmed that the noise of signal mainly exists in the high frequency component. In this paper, the IMF1 is a high frequency component by modal decomposition. Thus, the threshold q is set to 0.85 according to the ETC of IMFs.

Step 2: We suppose that the RCMDE of the remaining IMFs has a value range of [m,n], the average value is w. Take two thresholds in [m,n], namely $w_{1}$ and $w_{2}$($w_{1}<w_{2}$), these IMFs are divided into three parts, namely $X\left(m\leq RCMDE<w_{1}\right)$, $Y\left(w_{1}\leq RCMDE\leq w_{2}\right)$, $Z\left(w_{2}<RCMDE\leq n\right)$.

Step 3: Firstly, the $w_{1}$ is determined to be a fixed value. And then, the value of $w_{2}$ is continuously adjusted until the signal-to-noise ratio (SNR) of the reconstructed sequences Z is the largest and the root mean square error (RMSE) of Z is the smallest. Finally, the RCMDE of $w_{2}$ is defined as the final value of q.

Step 4: The value of $w_{1}$ is continuously adjusted until the SNR of the reconstructed sequences X is the largest and the RMSE of X is the smallest according to the algorithm of Step 3. Therefore, the RCMDE of $w_{1}$ is can be defined as the final value of r. After repeated experiments, the values of q and r are roughly determined as 1.85 and 1.10.

### 3.2. Evaluation Method of Chaotic Time Series

In order to evaluate chaotic time series more conveniently, many scholars have proposed some evaluation methods, such as signal-to-noise ratio (SNR), root mean square error (RMSE), correlation dimension, Lyapunov exponent and noise intensity [41,42] and so forth. Thus, in this paper, these evaluation methods are used to evaluate the effect of the proposed noise reduction method.

#### 3.2.1. Signal-To-Noise Ratio (SNR)

Signal-to-noise ratio shows an energy relationship between signal and noise. The higher SNR, the more useful information and the less noise of signal. Therefore, the SNR is a very intuitive method to evaluate the effect of denoised signal by analyzing whether the SNR is improved. The definition of SNR is defined as follows:(21)$$SNR=10\cdot \log_{10}\left(\frac{{\left\|x\right\|}^{2}}{{\left\|\hat{x}-x\right\|}^{2}}\right)$$
where x, $\hat{x}$ and ‖∗‖ respectively indicate the noise signal, denoised signal and norm. 

#### 3.2.2. Root Mean Square Error (RMSE)

Root mean square error shows the difference between denoised signal and original signal in numerical. and the smaller RMSE, the better noise reduction effect. The RMSE is defined as follows:(22)$$RMSE=\sqrt{\frac{{\left\|\hat{x}-x\right\|}^{2}}{length(x)}}$$
where length(∗) represents the length of signal. 

#### 3.2.3. Correlation Dimension

Fractal dimension is an important parameter to quantitatively analyze the chaotic attractor, which is applied to describe the nonlinear behavior of system. Correlation dimension is a branch of fractal dimension, it has been widely used in signal processing because of simple calculation. In 1983, Grassberger and Procacca proposed the GP algorithm for calculating the correlation dimension of time series [43]. For the time series $\left\{x_{1},x_{2},\ldots ,x_{n}\right\}$, let the embedding dimension of reconstructed phase space is m, the delayed sampling is applied to a series with a delay τ. The reconstructed phase space is as follows:(23)$$\left(\begin{array}{cccc} x_{1} & x_{2} & \cdots & x_{N}\\ x_{1+\tau } & x_{2+\tau } & \cdots & x_{N+\tau }\\ \vdots & \vdots & \ddots & \vdots \\ x_{1+\left(m-1\right)\tau } & x_{2+\left(m-1\right)\tau } & \cdots & x_{N+\left(m-1\right)\tau } \end{array}\right)$$
where N=n−(m−1)τ. For the reconstructed dynamical system, strange attractors are composed of $y_{i}=\left(x_{i},x_{i+\tau },x_{i+2\tau },\ldots ,x_{i+\left(m-1\right)\tau }\right)$. For any two vectors of $y_{i}$ and $y_{j}$ in phase space, the distance between them are as follow:(24)$$\left|y_{i}-y_{j}\right|=\underset{1\leq k\leq m}{\max}\left|y_{ik}-y_{jk}\right|$$

Suppose there are N vectors in the reconstructed phase space, the correlation integral is defined as:(25)$$C\left(r\right)=\frac{1}{N^{2}}\left(\sum_{i,j}H\left(r-\left|y_{i}-y_{j}\right|\right)\right)$$
where H(x) is Heaviside unit function.
(26)$$H\left(x\right)=\left\{ \begin{array}{l} 0,x\leq 0\\ 1,x>0 \end{array}\right.$$

The correlation integral C(r) has the following relationship with r when r→0:(27)$$\underset{r\to 0}{\lim}C_{n}\left(r\right)\propto r^{D}$$
where D represents correlation dimension, it can be obtained by calculating $D=\frac{\ln\left(C\left(r\right)\right)}{\ln\left(r\right)}$. 

#### 3.2.4. Noise Intensity

For the time series {x(n)}, whose noise intensity is approximated by standard deviation σ.
(28)$$\sigma =\sqrt{\frac{1}{N}\sum_{n=1}^{N}{\left[x\left(n\right)-\bar{x}\right]}^{2}}$$
where n=0,1,…,N, $\bar{x}=\frac{1}{N}\sum_{n=1}^{N}x\left(n\right)$ represents the mean of time series, it is found that the smaller noise intensity, the better the noise reduction effect.

#### 3.2.5. Lyapunov Exponent

The Lyapunov exponent judges the chaotic characteristics of the system based on the presence or absence of diffusion motion characteristics of the phase trajectory. Lyapunov exponent is defined as:(29)$$\lambda =\underset{n=\infty }{\lim}\frac{1}{n}\sum_{i=0}^{n-1}\ln{\left|\frac{df\left(x\right)}{dx}\right|}_{x=x_{i}}$$
where *n* represents the number of iterations, f(x) is the differential equation of the dynamic system and *x* is the distance of the neighboring points. The positive and negative magnitudes of Lyapunov $\lambda_{i}$ respectively represent the degree of divergence or convergence of adjacent trajectories in the phase space. Normally, we only need to calculate the maximum Lyapunov exponent. If the maximum Lyapunov exponent is a positive number, we can determine that there is chaotic component in the system. In this paper, the maximum Lyapunov exponent is used to quantitatively analyze the phase space attractors of the signals.

## 4. The Chaotic Signal Denoising Experiment

In this section, the Chens model is selected for simulation experiments and added Gaussian white noise with different SNR as input signals. In order to verify the noise reduction effect of the proposed algorithm, two combined noise reduction methods are chosen to compare with CEEMDAN_ETC_RCMDE_WSTD. They are EMD_MSE_WSTD and EEMD_DE_WSTD, the first two methods divide IMFs into two reconstructed series, namely noise-dominant IMFs, real signal-dominant IMFs. and we can obtain the final denoised signal by combining real signal-dominant IMFs. The Chens system is expressed as:(30)$$\left\{ \begin{array}{l} \hat{x}=a\left(y-x\right)\\ \hat{y}=\left(c-a\right)x-xz+cy\\ \hat{z}=xy-bz \end{array}\right.$$
where a=35, b=3, c=28. The equation is integrated by using a fourth-order Runge–Kutta method with a fixed step size of 0.01 and the initial value of the equation are x(0)=0, y(0)=1, z(0)=0. The *x* component signal with a length of 2048 points is selected as the chaotic signal, the Chens signal are added Gaussian white noise with different SNR. The denoised results of Chens signals with SNR are −10 dB, 0 dB, 10 dB and 20 dB are shown in Table 1. The time-domain waveform and phase space attractors of noisy Chens signal with 10dB are shown in Figure 4 and Figure 5.

It can be seen from Table 1 that the results of the denoised signals are improved by the three methods. However, the SNR of the proposed algorithm is higher, the RMSE is lower. As shown in Figure 4, the clarity and similarity of time-domain waveform by the proposed algorithm are the highest. Which not only achieves noise reduction but also restores the most of useful information. In Figure 5, although the three methods reduce the noise interference on a certain degree, the geometry of attractor obtained by the proposed algorithm has stronger regularity and higher clarity. In order to quantitatively analyze the phase space attractors of the Chens signal before and after noise reduction, the maximum Lyapunov exponent, correlation dimension and noise intensity before and after noise reduction can be calculated. The Characteristic parameters before and after noise reduction for Chens signal with SNR is 10 dB are shown in Table 2.

It can be seen from Table 2 that after the above three methods are used to denoise the Chens signal, the above characteristic parameters of the chens signal are significantly improved compared with before the noise reduction, the improvement of the CEEMDAN_ETC_RCMDE_WSTD is most obvious. Thus, it shows that the noise reduction effect of the proposed algorithm is better than the other two methods. 

## 5. The Underwater Acoustic Signals Denoising Experiment

In order to further verify the effectiveness of this proposed algorithm for chaotic signals. The data used in this paper are three different types of real underwater acoustic signals measured by calibrated omnidirectional hydrophone in the south China sea, namely the Ship-1, Ship-2 and Ship-3. Each type of underwater acoustic signals has 100 sample data. Each sample length is 2048 points and sampling interval is 0.05 ms. The sample data have been filtered, normalized and sampled before the noise reduction experiment, the decomposition results by CEEMDAN are shown in Figure 6. 

It can be seen from Figure 6 that the three types of underwater acoustic signals are decomposed into several IMFs. The different time scale components are not included in single IMF, the same scale component does not appear in different IMFs. It shows that CEEMDAN does not exhibit mode mixing and boundary effects when applied to underwater acoustic signals, which will make more sense for subsequent processing of underwater acoustic signals. The reconstructed series of underwater acoustic signals are shown in Table 3.

It can be seen from Table 3 that Ship-1, Ship-2 and Ship-3 are respectively divided into four parts. There is no problem that an IMF is repeatedly defined or undefined. It shows that the proposed algorithm meets the requirements of the IMF. Which will greatly contribute to the noise reduction of the underwater acoustic signals. The time-domain waveform of underwater acoustic signals and phase space attractors after noise reduction are respectively shown in Figure 7, Figure 8 and Figure 9.

As shown in Figure 7a, Figure 8a and Figure 9a, the time domain waveform before noise reduction is full of noise, some useful information and the change of time-domain waveform cannot be distinguished. It can be seen from Figure 7b, Figure 8b and Figure 9b that the noise of the underwater acoustic signals are well suppressed and the waveform change of the denoised signals are clearer by comparing 300 points before and after noise reduction. In addition, we can also determine whether the noise is effectively removed by comparing the chaotic attractors of the underwater acoustic signals before and after noise reduction. Because the degree of damage of the attractor self-similar structure is determined by the noise intensity. The greater the noise of signal, the weaker the regularity of attractor trajectory and the lower the self-similarity. It can be seen from Figure 7c,d, Figure 8c,d and Figure 9c,d that the regularity of denoised signals are stronger, the self-similarity are higher. It shows that the proposed algorithm can reduce the noise interference to a large extent.

In order to further quantitatively describe the effectiveness of the proposed algorithm by calculating the correlation dimension, noise intensity, PE and RCMDE for noise signals and denoised signals. The results before and after noise reduction are shown in Table 4. 

As shown in Table 4, the change of correlation dimension, noise intensity, PE and RCMDE are smaller than original signals. However, the change of CEEMDAN_ETC_RCMDE_WSTD is the most obvious, which indicates that the noise is greatly suppressed, the complexity is greatly reduced, Therefore, it is shown that the proposed algorithm can not only effectively remove most of the noise also the chaotic characteristics of underwater acoustic signals is greatly improved. Which will have great advantages in processing actual underwater acoustic signals. 

## 6. Conclusions

In order to solve the problem that inaccurate discrimination of IMFs because of imperfect decomposition process of EMD denoising algorithm and poor self-adaptability, a noise reduction method of underwater acoustic signal denoising based on CEEMDAN, combining ETC, RCMDE and wavelet threshold denoising is proposed. The innovations and conclusions of the proposed denoising method are as follows:

(1) CEEMDAN, as an adaptive decomposition algorithm based on EEMD, is introduced for underwater acoustic signal denoising. which has great development potential in the field of non-linear signal processing.

(2) Compared with existing denoising methods, the IMFs by CEEMDAN are divided into four parts (noise IMFs, noise-dominant IMFs, real signal-dominant IMFs and real IMFs) for the first time.

(3) The RCMDE is better than MSE and DE in analyzing the complexity of chaotic signals, is introduced for underwater acoustic signal denoising. Thus, the RCMDE will have greater potential in chaotic signals processing.

(4) The proposed method is applied to Chens model and three different types of real underwater acoustic signals. The proposed method is compared with EMD_MSE_WSTD and EEMD_DE_WSTD, making qualitative and quantitative analysis for denoised signals. The results show that the proposed algorithm can not only reduce the noise interference to a large extent also obtain more regular and clear chaotic attractors. Which will play an important role in researching the physical characteristics of underwater acoustic signals based on chaos theory.

## Figures and Tables

**Figure 1 entropy-21-00011-f001:**
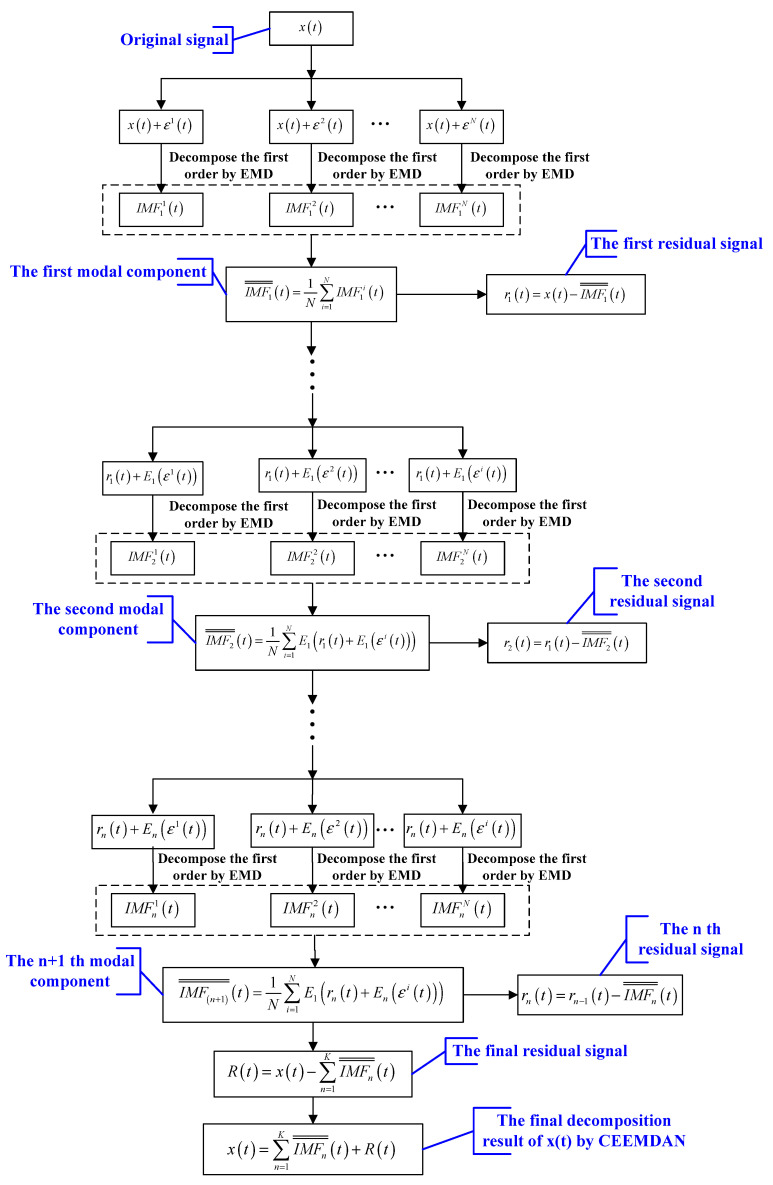
The flow chart of CEEMDAN.

**Figure 2 entropy-21-00011-f002:**
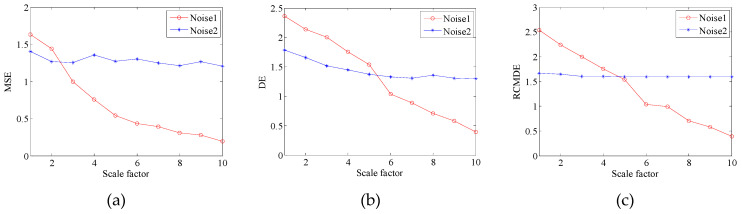
Mean value and standard deviation of results of the Noise 1 and Noise 2. (**a**) MSE; (**b**) DE; (**c**) RCMDE.

**Figure 3 entropy-21-00011-f003:**
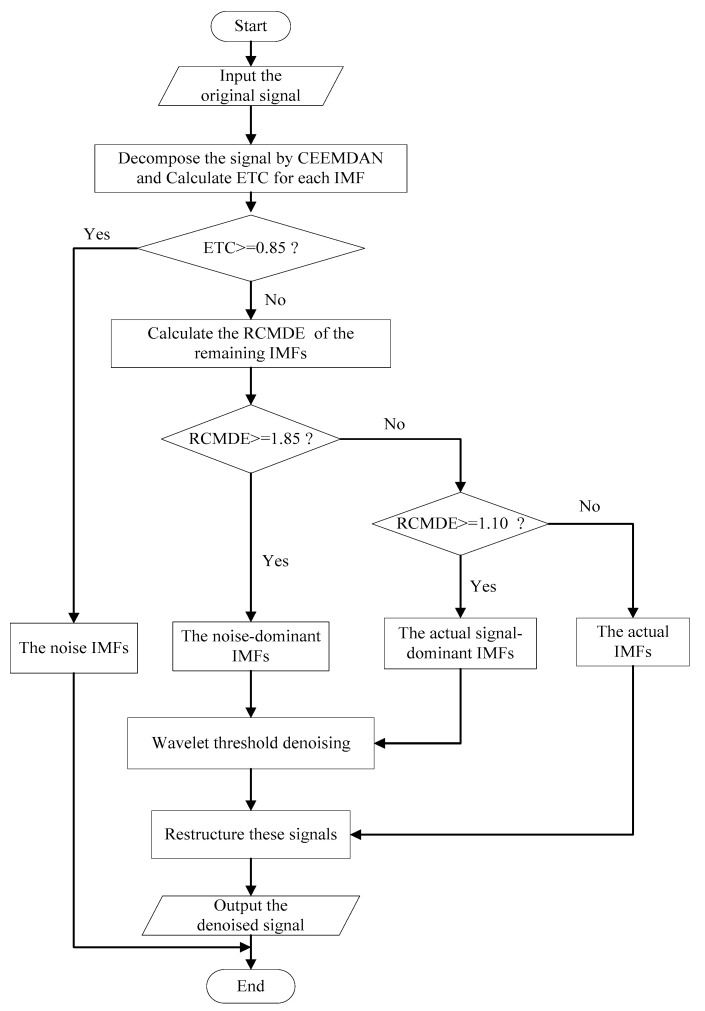
The flow chart of the proposed algorithm.

**Figure 4 entropy-21-00011-f004:**
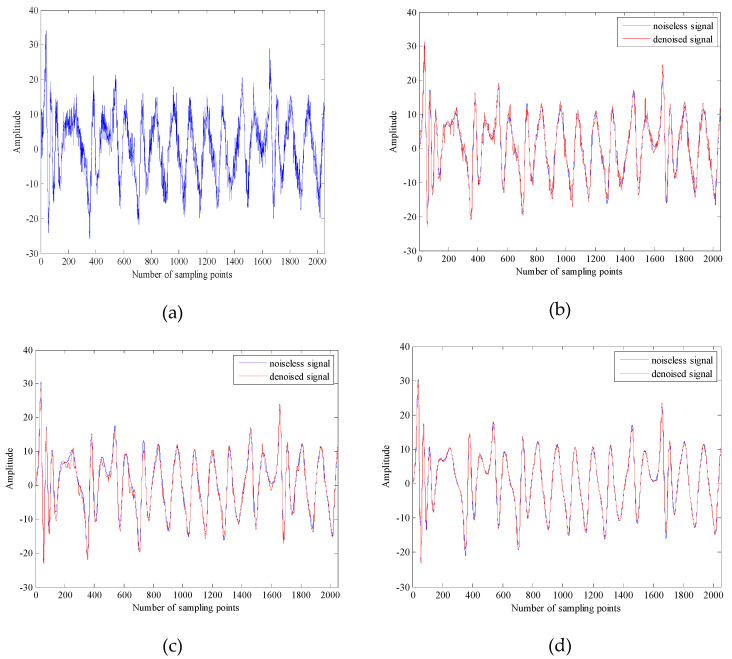
The time domain waveform before and after noise reduction of Chens signal with SNR is 10 dB. (**a**) The time-domain waveform of the noisy Chens signal with 10dB; (**b**) The time-domain waveform after noise reduction by EMD_MSE_WSTD; (**c**) The time-domain waveform after noise reduction by EEMD_DE_WSTD; (**d**) The time-domain waveform after noise reduction by the proposed algorithm.

**Figure 5 entropy-21-00011-f005:**
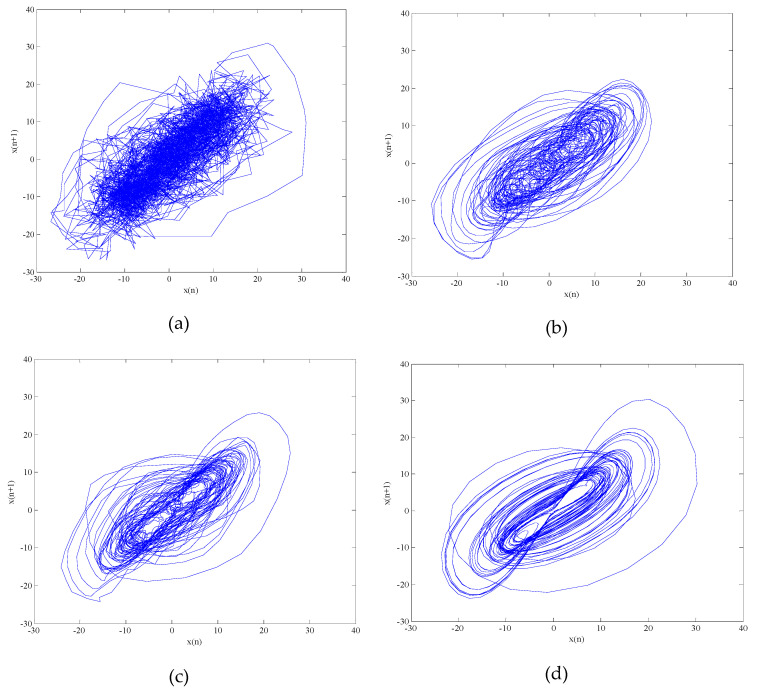
The phase space attractors before and after noise reduction for Chens signal with SNR is 10 dB. (**a**) The phase space attractor of the noisy Chens signal with 10dB; (**b**) The phase space attractor after noise reduction by EMD_MSE_WSTD; (**c**) The phase space attractor after noise reduction by EEMD_DE_WSTD; (**d**) The phase space attractor after noise reduction by the proposed algorithm.

**Figure 6 entropy-21-00011-f006:**
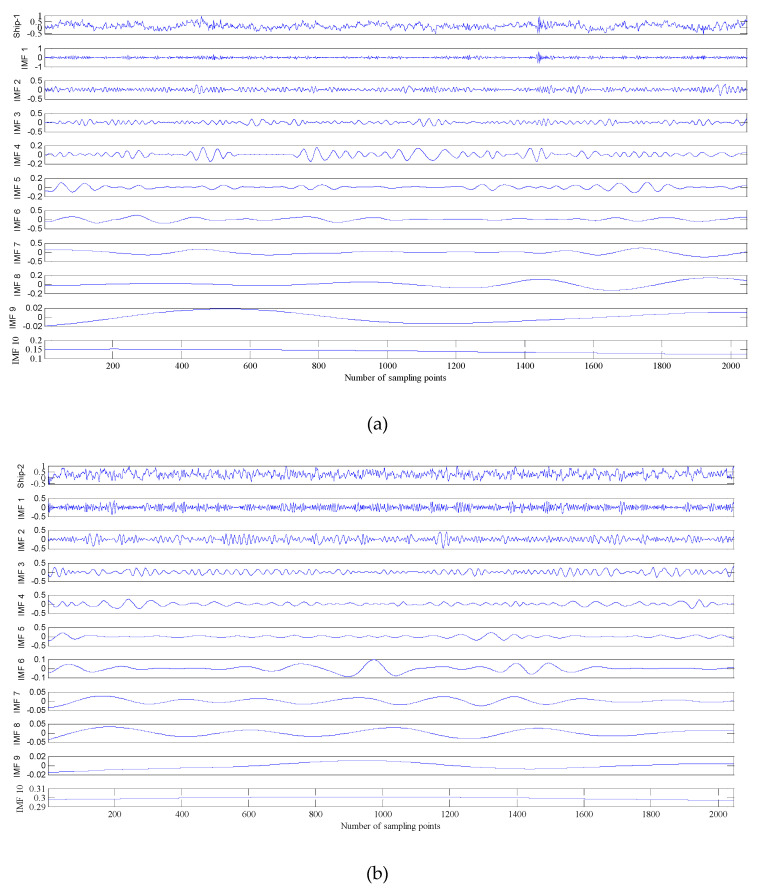
The decomposition result of underwater acoustic signals by CEEMADN. (**a**) The decomposition result of Ship-1; (**b**) The decomposition result of Ship-2; (**c**) The decomposition result of Ship-3.

**Figure 7 entropy-21-00011-f007:**
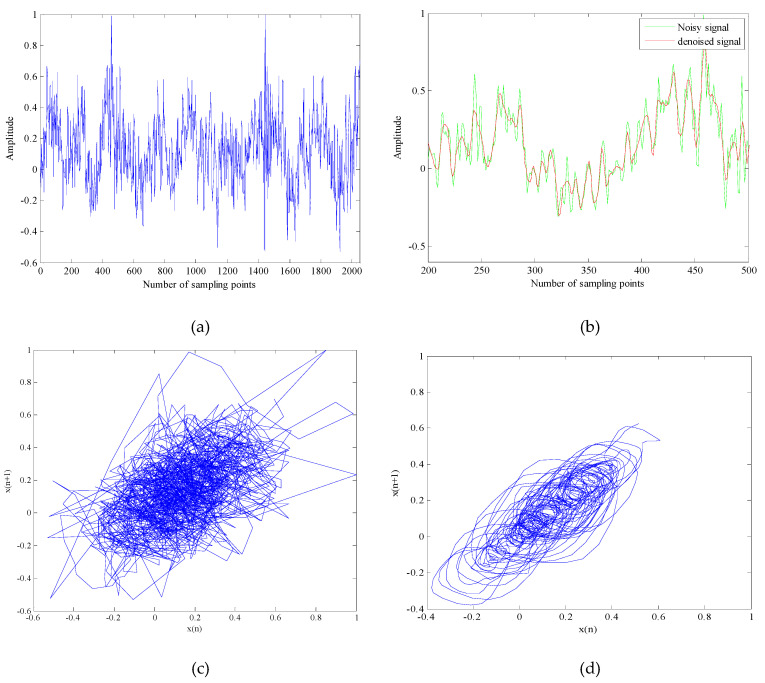
The time-domain waveform and phase space attractors of Ship-1 and denoised Ship-1. (**a**) The time-domain waveform of Ship-1; (**b**) The time-domain waveform of denoised Ship-1 with 300 points; (**c**) The phase space attractors of Ship-1; (**d**) The phase space attractors of denoised Ship-1.

**Figure 8 entropy-21-00011-f008:**
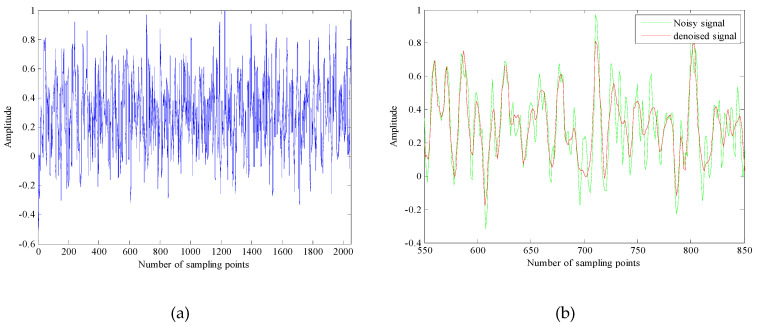
The time-domain waveform and phase space attractors of Ship-2 and denoised Ship-2. (**a**) The time-domain waveform of Ship-2; (**b**) The time-domain waveform of denoised Ship-2 with 300 points; (**c**) The phase space attractors of Ship-2; (**d**) The phase space attractors of denoised Ship-2.

**Figure 9 entropy-21-00011-f009:**
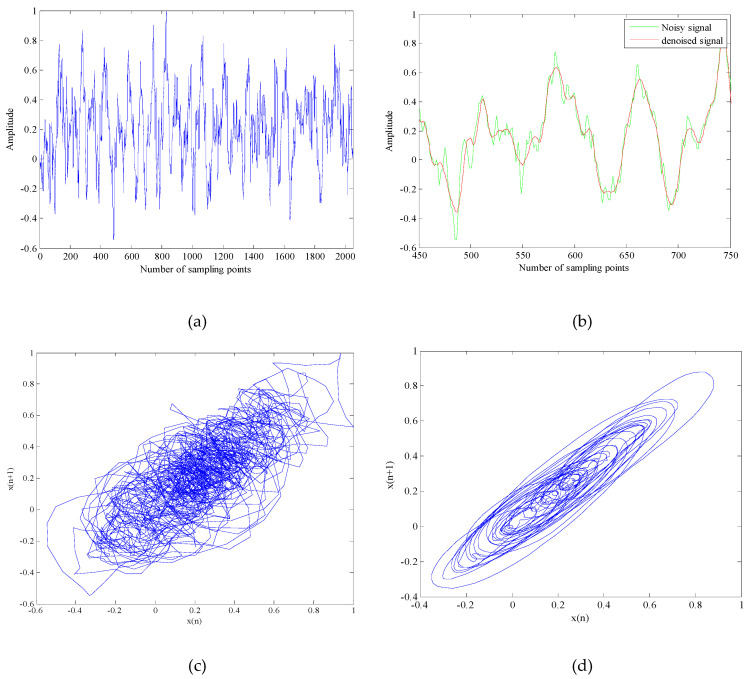
The time-domain waveform and phase space attractors of Ship-3 and denoised Ship-3. (**a**) The time-domain waveform of Ship-3; (**b**) The time-domain waveform of denoised Ship-3 with 300 points; (**c**) The phase space attractors of Ship-3; (**d**) The phase space attractors of denoised Ship-3.

**Table 1 entropy-21-00011-t001:** Denoising results of Chens signals.

SNR/dB	EMD_MSE_WSTD		EEMD_DE_WSTD		CEEMDAN_ETC_RCMDE_WSTD	
	SNR/dB	RMSE	SNR/dB	RMSE	SNR/dB	RMSE
−10	2.4455	0.9591	3.7687	0.7750	4.4430	0.6691
0	8.5354	0.7234	11.0066	0.5883	13.1377	0.4850
10	16.3106	0.4117	18.8902	0.3119	22.1273	0.2556
20	24.7166	0.2448	26.5438	0.1756	31.4315	0.0807

**Table 2 entropy-21-00011-t002:** The Characteristic parameters before and after noise reduction for Chens signal.

Signals	Maximum Lyapunov Exponent	Correlation Dimension	Noise Intensity
Chens signal (SNR = 10 dB)	0.3487	2.4369	0.2538
EMD_MSE_WSTD	0.2973	2.1702	0.2071
EEMD_DE_WSTD	0.2142	1.8960	0.1875
CEEMDAN_ETC_RCMDE_WSTD	0.1418	1.4542	0.1442

**Table 3 entropy-21-00011-t003:** The reconstruction series of underwater acoustic signals.

Underwater Acoustic Signals	Noise IMFs	Noise-Dominant IMFs	Real Signal-Dominant IMFs	Real IMFs
Ship-1	IMF1	IMF2, IMF3	IMF4, IMF5, IMF6, IMF7	IMF8, IMF9, IMF10
Ship-2	IMF1	IMF2, IMF3, IMF4	IMF5, IMF6, IMF7, IMF8	IMF9, IMF10
Ship-3	IMF1	IMF2, IMF3	IMF4, IMF5, IMF6	IMF7, IMF8, IMF9, IMF10

**Table 4 entropy-21-00011-t004:** The results before and after noise reduction.

Underwater Acoustic Signals	Parameters	Before Noise Reduction	EMD_MSE_WSTD	EEMD_DE_WSTD	CEEMDAN_ETC_RCMDE_WSTD
Ship-1	Correlation Dimension	1.7821	1.6698	1.5356	1.4906
	Noise Intensity	0.2250	0.1401	0.0977	0.0703
	PE	1.5525	1.0854	0.8366	0.6758
	RCMDE	2.5232	1.2350	0.9800	0.4492
Ship-2	Correlation Dimension	1.6751	1.5405	1.3438	1.2004
	Noise Intensity	0.2354	0.1380	0.1037	0.0893
	PE	1.6001	0.9055	0.7686	0.5359
	RCMDE	2.3280	1.2022	0.6991	0.3658
Ship-3	Correlation Dimension	1.6869	1.4772	1.2355	1.1890
	Noise Intensity	0.2512	0.1545	0.1221	0.0927
	PE	1.5466	0.9495	0.6792	0.5004
	RCMDE	2.1359	1.0304	0.6751	0.3047

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
