# Peer review of "Noise Reduction Method of Underwater Acoustic Signals Based on CEEMDAN, Effort-To-Compress Complexity, Refined Composite Multiscale Dispersion Entropy and Wavelet Threshold Denoising"

_entropy, 2018, doi:10.3390/e21010011_

Round 1

Reviewer 1 Report

The publication entitled “A New Underwater Acoustic Signals Denoising Technique Based on CEEMDAN, Effort-To-Compress Complexity, Refined Composite Multiscale Dispersion Entropy and Wavelet Threshold Denoising”, which has been sent for review, presents interesting research problem. This research present an interesting direction of the authors' research development, especially in the field of complex signal processing and it application.

In reviewer opinion the manuscript should be corrected:

·         chapter 1 - the authors should clearly indicate their previous achievements in signal processing

·         chapter 2.1 - line 87 - vocabulary error

·         line 224 - vocabulary error

·         chapter 2.3 - supplement with examples where the methods were used, give examples of the implementation of these selection methods from other works

·         line 225-235 - sort the numbering

·         chapter 5 - unambiguously compare the results of tests with the results obtained in previous studies

·         lack in reference and in the introduction chapter the state of the art of using denoising of signal, decomposition of signals and entropy  - for example:

-          An, X.; Yang, J. Denoising of hydropower unit vibration signal based on variational mode decomposition and approximate entropy. Trans. Inst. Meas. Control 2016, 38, 282–292

-          Wu, Haiyan; Chen, Yu; Lin, Weiguo Novel Signal Denoising Approach for Acoustic Leak Detection JOURNAL OF PIPELINE SYSTEMS ENGINEERING AND PRACTICE  Volume: 9   Issue: 4      2018

-          Xiao, Maohua; Wen, Kai; Zhang, Cunyi; Research on Fault Feature Extraction Method of Rolling Bearing Based on NMD and Wavelet Threshold Denoising SHOCK AND VIBRATION, 2018

-          Figlus T., J. Gnap, T. Skrucany, B. Sarkan, J. Stoklosa. 2016. The Use of Denoising and Analysis of the Acoustic Signal Entropy in Diagnosing Engine Valve Clearance. Entropy 18(7): 1-11

-          Geng, Z.; Chen, J. Investigation into piston-slap-induced vibration for engine condition simulation and monitoring. J. Sound Vib. 2005, 282, 735–775

-          Bi, F.; Li, L.; Zhang, J.; Ma, T. Source identification of gasoline engine noise based on continuous wavelet transform and EEMD-RobustICA. Appl. Acoust. 2015, 100, 34–42

-          Li, N.; Yang, J.; Zhou, R.; Liang, C. Determination of knock characteristics in spark ignition engines: An approach based on ensemble empirical mode decomposition. Meas. Sci. Technol. 2016, 27, 045109

-          Wang, Jianlin; Wei, Qingxuan; Zhao, Liqiang; An improved empirical mode decition method using second generation wavelets interpolation DIGITAL SIGNAL PROCESSING  Volume: 79   Pages: 164-174    AUG 2018

-          Figlus, Tomasz; Stanczyk, Marcin; Diagnosis of the wear of gears in the gearbox using the wavelet packet transform; METALURGIJA, Volume: 53  Issue: 4  Pages: 673-676, 2014

-          Thirumalaisamy, Mruthun R.; Ansell, Phillip J. Fast and Adaptive Empirical Mode Decomposition for Multidimensional, Multivariate Signals IEEE SIGNAL PROCESSING LETTERS  Volume: 25   Issue: 10   Pages: 1550-1554   OCT 2018

Conclusion: this manuscript require the correction and after that should be presented in Entropy.

Regards

Author Response

Review comments are available in the attachment.

Reviewer 2 Report

The paper addresses the problems related to imperfect decomposition process of empirical mode

decomposition (EMD) based denoising algorithm and proposes a new technique for underwater

acoustic signal denoising based on complete ensemble empirical mode decomposition with

adaptive noise (CEEMDAN). The paper is technically sound and addresses a problem that is relevant in many domains of application, where signal processing techniques are used to deal with noise contamination. I feel the paper would be of a great interest to the journal readers.

Comments:

-          the title of the paper is over-complicated and not clear; please shorten while avoiding the generic terms such as „new“

-          it is not clear, how the values of p, q and r in the algorithm description on p.10 were selected

-          Figure 5 only provides visual comparison of attractors in phase space: please provide a numerical comparison before and after denoising using, e.g., Attractor Quality Index (AQI).

-          „the geometry of attractor obtained by the proposed algorithm has stronger regularity and higher clarity“ -> provide numerical evaluation of attractor regularity;

-          Provide the evaluation of computational complexity of your method and compare it with complexity of classical EMD and its extensions;

-          Conclusions: rewrite, be more specific in summarizing the results achieved; discuss advantages and limitations of your method, and formulate implications for application in other domains as well. Specifically, you need to discuss how your method deals with well known problems of EMD such as mode mixing, boundary problem, and the physical meaning of IMFs.

-          Several important recent references are missing in the overview of related work, see:

·         IMF mode demixing in EMD for jitter analysis. Journal of Computational Science, 22, 240-252. doi:10.1016/j.jocs.2017.04.008

·         Nonnegative matrix factorization based decomposition for time series modelling doi:10.1007/978-3-319-59105-6_52

·         Emotion recognition using empirical mode decomposition and approximation entropy. Computers and Electrical Engineering, 72, 383-392. doi:10.1016/j.compeleceng.2018.09.022

·         Fractional empirical mode decomposition energy entropy based on segmentation and its application to the electrocardiograph signal. Nonlinear Dynamics, 94(3), 1669-1687. doi:10.1007/s11071-018-4448-y

Author Response

(The authors gave the same response as above.)

Reviewer 3 Report

This manuscript presents a technique for underwater acoustic signal denoising based on complete ensemble empirical mode decomposition with adaptive noise (CEEMDAN), effort-to-compress, refined composite multiscale dispersion entropy and wavelet threshold denoising. However, this manuscript must be significantly improved based on the following comments.

The title of the manuscrip must be changed. This title is confusing and large. In addition, all the contain of the manuscript is cofusing. This manuscript requires extensive editing of English language and style. The manuscript must include the main innovation or scientific contribution with respect to other techniquesreported in the literature. Authors must mention the advantages and challenges of their proposed techniques. All the sections must be modified. A manuscript must contain the sections  of introduction, methods and materias, results and discussion and conclusions.  This manuscript does not contain the sections of methods, results and discussion. Authors need more discussions about the reported results. All the quality and size of the results reported in the Figures must be improved. Authors must mention more discussions about the results reported in the Figures 6-9. The size of the Figures 2, 6-9 must be improved. Authors must add the challenges or limitations of the proposed techniques. The conclusions must contain the research works in the future.

Author Response

(The authors gave the same response as above.)

Round 2

Reviewer 3 Report

Authors have improved their manuscript based on all the reviewer's comments. They developed a technique for underwater acoustic signal denoising, which is based on complete ensemble empirical mode descomposition with adaptive noise (CEEMDAN), effort-to-compress complexity (ETC), refined composite multiscale dispersion entropy (RCMDE) and wavelet threshold denoising. This technique is used to noise reduction of real underwater acoustic signals.